# Psoriasis and Cardiometabolic Diseases: Shared Genetic and Molecular Pathways

**DOI:** 10.3390/ijms23169063

**Published:** 2022-08-13

**Authors:** Stefano Piaserico, Gloria Orlando, Francesco Messina

**Affiliations:** Unit of Dermatology, Department of Medicine, University of Padua, Via V. Gallucci 4, 35128 Padua, Italy

**Keywords:** psoriasis, cardiovascular, MACE, myocardial infarction, atherosclerosis, hypertension, hyperlipidemia, diabetes, obesity

## Abstract

A convincing deal of evidence supports the fact that severe psoriasis is associated with cardiovascular diseases. However, the precise underlying mechanisms linking psoriasis and cardiovascular diseases are not well defined. Psoriasis shares common pathophysiologic mechanisms with atherosclerosis and cardiovascular (CV) risk factors. In particular, polymorphism in the IL-23R and IL-23 genes, as well as other genes involved in lipid and fatty-acid metabolism, renin–angiotensin system and endothelial function, have been described in patients with psoriasis and with cardiovascular risk factors. Moreover, systemic inflammation in patients with psoriasis, including elevated serum proinflammatory cytokines (e.g., TNF-α, IL-17, and IL-23) may contribute to an increased risk of atherosclerosis, hypertension, alteration of serum lipid composition, and insulin resistance. The nonlinear and intricate interplay among various factors, impacting the molecular pathways in different cell types, probably contributes to the development of psoriasis and cardiovascular disease (CVD). Future research should, therefore, aim to fully unravel shared and differential molecular pathways underpinning the association between psoriasis and CVD.

## 1. Introduction

Psoriasis is a chronic skin immune-mediated inflammatory disease with a global prevalence of 1–3%; it presents as raised, erythematous, and scaly plaques [1]. Psoriasis-related inflammation may extend beyond the skin [2]. Patients with psoriasis have an increased risk for cardiovascular (CV) disease (CVD), a common cause of morbidity and mortality in psoriasis, and CV risk factors [3,4,5]. In particular, an independent association between psoriasis and an increased risk of myocardial infarction (MI), chronic heart failure (CHF), and cardiac arrythmia has been demonstrated [6,7,8,9]. In this paper, we focused our attention on ischemic heart disease (IHD). IHD describes a group of clinical syndromes characterized by myocardial ischemia, i.e., an imbalance between myocardial blood supply and demand [2]. Several mechanisms may lead to IHD. The most common is atherosclerotic coronary artery disease. However, a relevant proportion of patients may have no significant epicardial disease and often have microvascular disease as the underlying pathophysiology.

Notably, the increased ischemic CV risk in psoriasis patients seems to be associated with a higher activity of psoriasis. Indeed, severe psoriasis confers the highest CV risk (compared with control subjects), including up to threefold increased risk of MI, 60% increased risk of stroke, and 40% increased risk of CV-related deaths [6]. This association could only be partially explained by a higher prevalence of traditional modifiable CV risk factors [10,11,12,13,14,15,16,17] including hypertension, diabetes, hyperlipidemia, obesity, nonalcoholic fatty liver disease (NAFLD), and smoking, along with metabolic syndrome. These conditions are often neglected in patients with severe psoriasis [18]. In addition to this, chronic systemic inflammation associated with psoriasis is likely to independently contribute to the increased risk of IHD.

Here, we review current knowledge on the associations linking psoriasis, CV risk factors, and CV events, with an emphasis on the common molecular and genetic underpinnings.

## 2. Psoriasis as an Independent CV Risk Factor

### 2.1. Shared Genetic and Molecular Pathways between Psoriasis and Atherosclerosis

Several studies, involving large sample sizes and complex multivariate models, have consistently suggested that severe psoriasis may be an independent risk factor of CVD [4,19,20,21,22,23].

Cardiologists have, therefore, included it in the European and American guidelines on cardiovascular disease prevention as a 1.5 risk factor multiplier for CV risk [24], as well as a risk-enhancing factor for the start of statin treatment in nondiabetic adults with a CV intermediate risk [25].

Chronic psoriatic disease has a negative impact on vascular inflammation and major adverse cardiovascular events (MACE), suggesting that exposure to low-grade chronic inflammation may accelerate vascular disease development and MACE [26]. Vascular inflammation by 18-FDG PET/CT is significantly associated with disease duration (β = 0.171, *p* = 0.002). Moreover, the duration of psoriasis has a strong relationship with MACE even after accounting for age, sex, socioeconomic status, CVD history, smoking, alcohol abuse, diabetes, hypertension, and use of statins (an increase of 1.0% for each additional year of PSO, HR 1.010; 95% CI, 1.007 to 1.013).

Despite the epidemiological association between psoriasis and CVD, the precise underlying mechanisms linking psoriasis and cardiovascular diseases are not well defined. It is unknown whether localized, chronic cutaneous inflammation directly promotes vascular inflammation and atherosclerotic plaque formation.

Figure 1 illustrates pathogenesis of psoriasis, associated comorbidities and involved molecular pathways.

In psoriatic plaques, activation of plasmacytoid dendritic cells (pDCs) promotes the maturation of myeloid dendritic cells (mDCs) and the production of TNF-α, IL-12, and IL-23. This leads to the activation of T helper 1 and 17 (Th1 and Th17) cells and subsequent secretion of inflammatory cytokines, such as TNF-α, IL-17, IL-21, and IL-22 [30]. Keratinocytes are then activated by these cytokines, particularly IL-17, and they produce antimicrobial peptides, cytokines, and chemokines, contributing to the amplification of inflammation.

Interestingly, the pathophysiology of atherosclerosis and psoriasis shares common mechanisms.

Indeed, atherosclerosis and coronary artery disease are both chronic inflammatory disorders, involving the immune cells, including Th1 and Th17 cells, with inflammatory factor infiltration observed at all stages of the disease [2].

### 2.2. Genetic Background

No association between IL-12 or IL-17 gene polymorphism and CV diseases has been reported so far [31,32]. Instead, genetic studies have shown that polymorphisms of the IL-23 gene are associated with the occurrence or progression of coronary artery disease [32].

In particular, the IL-23R rs6682925T/C polymorphism appears to be linked to the development of coronary artery disease [33,34]. Moreover, two small studies have revealed an association between IL-23R and CVD. The functional single-nucleotide polymorphism (SNP) rs11209026 G/A in IL-23R, which is associated with reduced IL-23 receptor signal transmission, is described to be protective against atherosclerosis progression [35]. Moreover, the IL-23 gene rs2066808 polymorphism, already associated with susceptibility to psoriasis [36,37] and psoriatic arthritis [38], is reported to increase the genetic risk of premature coronary artery disease (CAD) [39]. Likewise, a large-scale genome-wide association study (GWAS) for psoriasis [40] showed that the variants of IL-23R are significantly associated with psoriasis susceptibility [41]. A TNF-α 238 G/A polymorphism was found to be significantly associated with a reduced risk of psoriasis in the general population [42].

On the contrary, TNF-α 238 G/A locus A has a significant association with CAD susceptibility in Europeans and North Asians [43]. Interestingly, transcriptional activity in people with the TNF-α-238A allele is significantly reduced, and monocytes carrying the TNF-α-238A allele in the peripheral blood, when stimulated, produce significantly less TNF-α [44].

### 2.3. Shared Molecular Pathways

Despite the high discrepancies between results obtained from different models, TNF-α has been recognized as one of the most potent proatherogenic cytokines [45], by increasing LDL transcytosis in endothelial cells [46,47] and by regulating the activity of macrophage scavenger receptor and foam cell formation [48]. Moreover, the blockade of TNF-α at the time of myocardial infarction improves cardiac functions, reduces infarct size, and promotes cardiomyocytes apoptosis [48].

TNF-α is also known to increase reactive oxygen species (ROS) levels and decrease nitric oxide production in blood vessels, which can lead to endothelial dysfunction, the initial step in the development of atherogenesis [49,50]. Similarly, several studies have consistently reported that the serum levels of TNF-α are significantly increased in patients with psoriasis compared to healthy controls [51].

Surprisingly, despite the central role of IL-12/23p40 and IL-17 in the pathogenesis of psoriasis, no significant difference in the serum levels of these molecules between the psoriasis and control groups was observed in most studies [51]. Indeed, serum cytokine levels do not necessarily reflect disease-specific activity, since their concentrations may be affected by several processes such as deposition and diffusion in other tissues. Nevertheless, IL-12 and IL-17 are produced in high amounts in psoriatic plaques, and several studies have reported a significant correlation between IL-17 and the psoriasis area and severity index (PASI) score [52,53,54]. Therefore, in patients with greater disease severity, skin-derived IL-12 and IL-17 may have a systemic impact.

Preclinical studies have suggested a possible role of IL-12 in the progression of atherosclerosis. Elevated serum IL-12 levels have been observed in ApoE-deficient mice, and they are associated with the progression of atherosclerosis [55]. Preclinical reports have also demonstrated that treatment with exogenous recombinant murine IL-12 significantly aggravated the progression of atherosclerosis, and it increased aortic atherosclerotic plaque areas in both ApoE-deficient mice and low-density lipoprotein receptor-deficient mice, whereas abrogating IL-12 levels significantly diminished such effects [56,57,58].

Plasma IL-12 concentrations have been demonstrated to be significantly higher in atherosclerosis and atherosclerotic cardiovascular disease [59], including stable angina pectoris, non-ST segment elevation MI, ST-elevation MI, and acute MI (reviewed in [32]). Most of these studies had small sample sizes and did not adjust for the multiplicity of confounding factors. Moreover, some studies excluded patients with autoimmune disorders. However, the consistency of these findings points toward the potential role of IL-12 in CV diseases.

IL-17 appears to play a less important role in atherosclerotic plaque development compared to Th1 cytokines (TNF-α and IL-12), and IL-17 confers both proatherogenic and protective effects in atherosclerosis, thus having controversial roles [60,61,62] (reviewed in [63]).

Th17 cells and IL-17 are present in murine and human atherosclerotic lesions [64]. IL-17 is increasingly expressed in symptomatic carotid plaques as opposed to asymptomatic plaques, and IL-17 mRNA expression appears higher in complex, unstable, and lipid-rich lesions [65]. In a murine model, IL-17A inhibition decreased the atheroma dimensions and the vessel stenosis, seemingly inducing a stabilization of the plaque, as shown by the increase in intravascular smooth muscle cells and in the collagen of the fibrous cap, the downregulation of metalloproteinases, and the decreased apoptosis in the lesion [66].

These findings are similar to those published in other studies in which recombinant IL-17A induced proatherogenic changes and increased plaque instability, whilst IL-17A inhibition reduced the development of atherosclerosis [64,67,68,69]. In another mouse model (K14-IL-17Aind/+) investigating IL-17A overexpression in keratinocytes inducing psoriasis-like lesions, evidence of vascular dysfunction and arterial hypertension, along with large aortic wall cellular infiltrates and reduced survival compared to the controls, was observed [70]. Likewise, Wang and al. showed that higher blood levels of IL-17 and TNF-α induced by cutaneous inflammation were associated with aortic inflammation and thrombosis [71]. These data sustain the view, at least in mouse models, that skin psoriasis-related inflammation promotes vascular inflammation and increases the risk of CVD.

Another contribution of IL-17 to the formation of atherosclerotic plaque seems to involve the induction of maturation and differentiation of macrophages; these cells, which are precursor of the foam cells, would then be activated by oxidized low-density lipoprotein (oxLDL), thereby starting the process of atherogenesis [72].

On the contrary, in another mouse model, a significant reduction in atherosclerotic lesion development was found in mice supplemented with rIL-17 [73]. A potential role of the IL-17-mediated mechanism of plaque stabilization has been suggested [69,74]. These contradicting observations may reflect the limitations of the experimental animal models of atherosclerosis used by the respective investigators. Overall, the majority of animal/preclinical data available to date suggest the role of IL-17 in the development of atherosclerosis or CVD.

In humans, higher IL-17 plasma levels were found in patients with coronary artery and carotid artery disease [75]. Moreover, high serum levels of IL-17A are associated with increased CV mortality [76]. This observation might, however, be interpreted both as a cause and as a result of compensatory mechanisms to ameliorate CV damaging effects.

In summary, the published evidence, while not definite, underpins the proatherogenic role of IL-17 and its involvement in the development of CV events. Taken together, these findings provide insight into skin-specific inflammatory contributions that support the epidemiological evidence indicating that patients with severe psoriasis have an increased risk of CVD and cardiovascular mortality.

## 3. Higher Prevalence of Traditional CV Risk Factors in Psoriasis Patients

Patients with psoriasis have an increased chance of traditional modifiable CV risk factors including hypertension, diabetes, hyperlipidemia, and obesity, all of which are combined and defined as metabolic syndrome [10,11,12,13,14,15,16,17]. Indeed, a “dose effect” of psoriasis on the metabolic syndrome and its components has been demonstrated [17]. Below, we discuss the genetic and molecular relationship between each component of metabolic syndrome and psoriasis. Table 1 summarizes the most relevant pathways shared between psoriasis and atherosclerosis or traditional CV risk factors.

### 3.1. Hypertension

The prevalence of hypertension in patients with psoriasis is estimated to be 38.8% [77], whereas it is about 31.1% in the general population [78]. Indeed, the risk of hypertension in patients with psoriasis was 1.7 times that of a control group [79], and the severity of psoriasis positively correlated with the risk of hypertension [80]. Although the association between psoriasis and hypertension has been documented, the mechanism underlying this link has not yet been unequivocally defined.

#### 3.1.1. Genetic Background

Numerous studies have been carried out to evaluate the genetic predisposition of patients with psoriasis to develop hypertension. Genetic studies, especially GWAS, found that there are some common genes/loci between psoriasis and its comorbidities.

Ogretmen et al. [81] explored genetic mutations of the *eNOS* gene coding for the endothelial nitric oxide enzyme in patients with psoriasis. This enzyme is involved in the production of nitric oxide at the endothelial level and, therefore, in the regulation of vascular smooth muscle tone. *eNOS* gene mutations are risk factors for coronary artery disease, myocardial infarction, and hypertension [82]. In particular, the researchers noticed a significantly increased frequency of the T allele in *eNOS* Glu298Asp in patients with psoriasis as compared to normotensive non-psoriatic healthy volunteers. The T allele frequency was also found to be greater in hypertensive psoriatic patients than in normotensive psoriatic patients. In conclusion, the results indicated that the Glu298Asp polymorphism of the *eNOS* gene appears to be an independent risk factor for psoriasis and is a potential genetic susceptibility for hypertension [81].

Cheng et al. [83] analyzed polymorphisms of the gene *LNPEP*, also named insulin-responsive aminopeptidase, a crucial component in the renin–angiotensin system pathway, and they identified it as an angiotensin IV receptor [84]. *LNPEP* and its genetic variants have been implicated in hypertension and diabetes [85,86] because of their biological effects on vasopressin clearance, serum sodium regulation [87], and glucose uptake via the interaction of insulin receptor signaling with the insulin-responsive glucose transporter GLUT4 [88]. In particular, researchers identified a missense *LNPEP* A763T mutation in patients with psoriasis leading to the disruption of peptide function, which was downregulated in the psoriatic skin compared with the control and uninvolved patient skin. Therefore, this raises the speculation that *LNPEP* may have a role in the pathogenesis of both psoriasis and metabolic conditions, such as hypertension and diabetes [83].

Interestingly, the angiotensin-converting enzyme (ACE) gene, which is also a key component in the renin–angiotensin system, was widely reported to be associated with psoriasis [89,90,91,92,93,94], further supporting the involvement of the renin–angiotensin system in the development of psoriasis and its related hypertension. In particular, different polymorphisms in the angiotensinogen gene (*AGT*) have been reported to influence the rate of *AGT* transcription [95], and the D allele is associated with higher concentrations of ACE mRNA in cells and increased ACE concentration and activity in plasma and serum [91]. Weger et al. demonstrated that *ACE* insertion/deletion(I/D) polymorphism may affect susceptibility to early-onset psoriasis [90]. Moreover, Veletza et al. showed that patients with psoriasis vulgaris or early-onset psoriasis have a statistically significantly higher frequency of I/D heterozygosity, as well as the presence of allele D compared to the controls [91].

#### 3.1.2. Molecular Pathways

Many pathophysiological and molecular pathways have been explored to shed light on the link between psoriasis and hypertension. Among these, the role of the autonomous nervous system (ANS), involved in the regulation of blood pressure and heart rate, has been explored, and evidence indicates that ANS is affected in patients with psoriasis [96,97]. Of note, Mastrolonardo et al. investigated the experimental stress-induced cardiovascular system response in patients with psoriasis. Post-stress systolic/diastolic blood pressure and hemodynamic responses of patients with psoriasis and those of the control group were compared, revealing statistically significantly increased values in patients with psoriasis.

Other evidence suggests that endothelin-1 may play a role in the development of hypertension among patients with psoriasis. Endothelin-1 is a mediator of vasoconstriction and increases blood pressure. It is synthetized by several cell types including keratinocytes, and its expression seems to be modified in psoriasis patients [98]. Compared to non-psoriatic controls, endothelin-1 expression appears to be higher in lesional skin and the blood of patients with psoriasis. Furthermore, endothelin-1 levels increase with psoriasis disease severity [98]. Higher endothelin-1 levels may, thus, contribute to the development of hypertension, due to an increased vasoconstrictive action on the blood vessels [99].

The enhancement of the renin–angiotensin signaling pathway may also induce hypertension less responsive to treatment in patients with psoriasis [80]. Renin is involved in the release of aldosterone, vasoconstriction, and increased blood pressure. Moreover, patients with psoriasis have an enhanced renin activity and increased urinary aldosterone excretion [100]. Interestingly, Suarez et al. performed a transcriptome analysis and observed increased expression of the renin gene in lesional skin of patients with moderate to severe psoriasis compared with matched non-lesional skin, suggesting that products of the psoriatic plaque have a hormone-like action and influence the biology of distal sites [101].

TNF-α contributes to an increased risk of hypertension and CHD through various mechanisms [102]. In particular, TNF-α blocks the activation of endothelial nitric oxide synthase (eNOS), degrades eNOS mRNA, alters vasomotor function by acting on vascular smooth muscle cells, and induces oxidative stress by increasing the production of ROS, leading to an unbalanced microvascular dilation/constriction [103,104,105]. Moreover, it has been hypothesized that crosstalk between the renin–angiotensin system and proinflammatory molecules, such as TNF-α, may regulate cardiovascular functions. Indeed, Sriramula et al. demonstrated the implications of TNF-α in mediating chronic angiotensin II-induced effects on increasing salt-appetite and blood pressure, possibly via its role in upregulating AT1 receptors, as well as enhancing NF-κB activity [106].

There is growing interest in the role of IL-17, which has been demonstrated to be overexpressed in patients with hypertension [107]. In a preclinical model of psoriasis, the dermal overexpression of IL-17A induces vascular oxidative stress and arterial hypertension [70]. Moreover, IL-17A also promotes endothelial dysfunction and angiotensin II-induced hypertension [108]. IL-17 may cause hypertension by inducing vessel inflammation and stiffness by inducing oxidative stress, as well as cardiac and renal fibrosis [109,110,111]. In line with the latter hypothesis, the administration of anti-IL-17 antibodies in a mouse model of hypertension reduced renal transforming growth factor-beta (TGF-β) levels, a well-known marker of fibrosis, compared to the control group. Inhibition of IL-17 signaling also ameliorated the progression of albuminuria and attenuated renal and vascular lymphocyte infiltration, thus suggesting that the inhibition of IL-17 may be a useful adjunct treatment for hypertension and the associated end-organ dysfunction [112].

### 3.2. Diabetes

Epidemiological studies strongly support an association between psoriasis and type 2 diabetes (T2D) [13,113,114,115]. Indeed, T2D is reported in 10–20% of patients with psoriasis [116] compared to a prevalence of 6.7% in the general population [117]. Of note, patients with psoriasis with normal blood glucose levels demonstrate insulin resistance or impaired insulin sensitivity, which may potentially result in the development of diabetes mellitus [118].

#### 3.2.1. Genetic Background

Genetic studies have suggested a shared mechanism between psoriasis and diabetes. Several genetic psoriasis susceptibility loci (PSORS) have been identified, and *PSORS2*, *PSORS3*, and *PSORS4* have been proposed as susceptibility loci of metabolic diseases, including T2D [113]. SNPs of the *CDKAL1* gene, which can reduce the glucose sensitivity of pancreatic beta-cells and, therefore, predispose individuals to the onset of diabetes, are also associated with a higher incidence of psoriasis [119,120]. Furthermore, some genetic polymorphisms in *JAZF1* and *ST6GAL1*, recently identified as causal risk genes of T2D in the general population [95], have been demonstrated to influence not only the risk for psoriasis but also disease severity [121]. Lastly, common DNA variants of IL-12B, IL-23R, and IL-23A were associated with increased susceptibility to psoriasis and contributed to an increased risk of developing T2D in a Spanish population [122].

#### 3.2.2. Molecular Pathways

Systemic inflammation in patients with psoriasis is increasingly thought to be the basis of the increased risk of T2D and insulin resistance [123]. Consistently, a “dose effect” of psoriasis severity on the risk of T2D has been reported [13,114], and insulin resistance significantly correlates with the PASI score [124]. TNF-α and IL-23/IL-17, as well as adipokines, have been proven to affect the regulation of insulin sensitivity by acting on the signaling pathways linking insulin receptors, cytokines, and adipocytokines [125,126]. In particular, TNF-α is known to impair insulin signaling via serine phosphorylation of IRS-1, thus leading to the reduction in GLUT-4 expression and a subsequent decrease in glucose entry into cells [127]. In line with this, in preclinical models, the exogenous administration of TNF-α induced insulin resistance, whereas animals deficient in TNF-α receptors were protected against insulin resistance [128,129,130]. Of note, treatment with the anti-TNF-α drugs etanercept and adalimumab, commonly used in the treatment of psoriasis, was shown to exert positive effects on insulin sensitivity [131,132]. Interestingly, patients with T2D, particularly female patients, show significantly higher IL-17 serum levels that correlate with age, insulin resistance, fasting blood sugar, BMI, and waist circumference [133]. Moreover, a reduction in glycated hemoglobin concentration (HbA1c) upon treatment in patients newly diagnosed with T2D has been associated with a decrease in IL-17 levels [134]. In animal models, it has been demonstrated that IL-17 deficiency is related to increased glucose tolerance and insulin sensitivity. Of note, IL-17 has also been associated with a reduction in serum insulin and is indicative of improved insulin sensitivity [135]. Intriguingly, in KK-Ay diabetic mice, the inhibition of IL-17 ameliorated insulin resistance and enhanced glucose uptake by muscle tissue, but not by fat tissue [136]. From a molecular point of view, it is reasonable to believe that IL-17 may be able to induce insulin resistance by acting on insulin signaling pathways [63]. In particular, it might inhibit the phosphorylation of IRS-1 via the direct activation of the IkB kinase (IKK)/NFkB pathway and via the indirect activation of JNK, through the induction of the expression of IL-6 [63,137,138].

Adipokines are a large family of cytokines including adiponectin, leptin, and resistin, which are generated and released by adipocytes and are involved in obesity and insulin resistance [139]. Adiponectin is inversely related to body mass index (BMI); hence, its expression and release decline with increasing obesity [140]. It enhances insulin sensitivity through activation of adenosine monophosphate protein kinase [141], thereby inhibiting acetyl-CoA carboxylase. As a result, it leads to a declined lipogenesis and is associated with enhanced fatty-acid beta-oxidation, and an increased internalization of glucose, through the expression of GLUT 1 and GLUT4. On the contrary, leptin, and resistin are considered insulin-antagonizing adipokines, as they inhibit insulin signaling pathways [142]. The adipokine milieu in patients with psoriasis is similar to that of prediabetic subjects [143], who present with lower levels of adiponectin [51,144,145] and increased levels of leptin and resistin compared to healthy controls [124,146]. Interestingly, it has been reported that both TNF-α and IL-17 may impact the production of adiponectin, resistin, and leptin by adipocytes [135,147,148].

### 3.3. Hyperlipidemia

The association between psoriasis and changes in blood lipid profile has been established by several studies [149], and dyslipidemia was found to be a risk factor for psoriasis development [150]. This association could be due to shared genetic variants, aberrantly activated molecular pathways, or lifestyle modifications [151].

#### 3.3.1. Genetic Background

The relevance of genetic predisposition for dyslipidemia in psoriasis was established by Mallbris et al., who demonstrated that lipid profile is altered at the initial stages of psoriasis development, suggesting that inflammatory actions follow the metabolic ones, and not vice versa [152].

In particular, novel SNPs in the *HLA* gene region, which is traditionally associated with immune-mediated conditions including psoriasis, correlate with increased risk for dyslipidemia [151].

*ApoE* polymorphisms, especially ε2 and ε4 in the European population, are associated with psoriasis onset and/or increased severity, while, on the other hand, the allele ε3 shows a protective effect [153]. At the same time, ApoE alleles, especially ε4, are associated with dyslipidemia [153].

Paraoxonase 1 (PON1) is an apolipoprotein-associated enzyme that protects lipoproteins from oxidative damage. Dysfunction of PON1 is implicated in metabolic disorders, and the *PON1* 55 M allele is a risk factor for psoriasis with an odds ratio of 1.57 for heterozygotes and 1.97 for homozygotes [154].

In psoriatic lesions, the expression levels of genes involved in lipid and fatty-acid metabolism are substantially altered; however, most of these local modifications do not have systemic consequences [155]. Conversely, a broader effect seems to be played by the genes *LXR-α* and *PPAR-α*, implicated in the metabolism of lipids. Notably, LXR-α regulates the transport of cholesterol from atherosclerotic lesions to the liver, while PPAR-α is a key lipid metabolism regulator, influencing Apo-A1 and HDL serum levels; thus, it is used as a therapeutic target in the treatment of dyslipidemia [156]. Both genes are downregulated in psoriatic skin, and the downregulation is proportional to skin inflammation. The downregulation of these genes also has systemic consequences due to the corresponding decline in Apo-A1 and HDL levels in blood [156].

#### 3.3.2. Molecular Pathways

The alteration of serum lipid composition is, at least partly, a consequence of chronic systemic inflammation. TNF-α is elevated in psoriasis and increases proatherogenic small dense LDL and ox-LDL levels while lowering HDL concentrations. Concomitantly, IL-6 and IL-1β induce VLDL synthesis and lower triglyceride clearance, further contributing to the production of small dense LDL and oxLDL [157]. Moreover, HDL isolated from psoriatic patients is less efficient in stimulating cholesterol efflux from macrophages, thus increasing cardiovascular risk despite normal HDL levels [158,159].

Furthermore, in several inflammatory disorders including psoriasis, the formation of anti-HDL antibodies has been reported. This correlates with increased (CV) risk, and the autoantibody titers in psoriasis seem to correlate with the clinical severity of the disease, providing a possible additional explanation for the increased CV risk reported in severe psoriasis [160].

However, dysfunctional HDL levels might be associated with immune dysregulation. In fact, by influencing the lipid structure of cell membranes, they interfere with immune cell physiology, modulate the maturation of DC cells, alter the expression of MHC molecules, play a role in lymphocyte activation, regulate cytokine production, and interfere with the complement system [161,162]. This might provide an alternative explanation to the observation that dyslipidemia occurs at an early stage of psoriasis development [152]. In addition to a possible common genetic background, a dysfunctional lipid profile could even contribute to the immune dysregulation underlying psoriasis.

Vitamin D deficiency is another common mechanism that could contribute to both psoriasis and dyslipidemia. Indeed, vitamin D contributes to the integrity of the skin barrier and modulates the functioning of the cutaneous immune system. Notably, it stimulates T regulatory cells, downregulates the expression of proinflammatory cytokines, and suppresses dendritic cell activation [163]. In keeping with this observation, vitamin D levels are lower in both psoriasis and psoriatic arthritis, compared to controls, and are inversely correlated with inflammatory markers and disease severity [164,165,166]. On the other hand, low levels of vitamin D have been associated with increased triglyceride levels in large epidemiologic studies [167,168]. Interventional studies showed that supplementation with vitamin D or its derivates results in a significant decrease in LDL and total cholesterol [169], as well as triglyceride levels [170,171,172]. It has, thus, been hypothesized that vitamin D might influence both intrahepatic metabolism and peripheral clearance of lipids through calcium and parathormone-mediated mechanisms [168], as well as modulate their synthesis by directly acting on gene transcription [169].

### 3.4. Obesity

Psoriasis and obesity are deeply intertwined conditions, as demonstrated by several epidemiologic studies. In fact, not only is psoriasis more common and more severe in patients with obesity, but obesity is also more prevalent in patients with psoriasis [12,17,173], including children [174]. The two conditions are linked by common lifestyle modifications such as unhealthy diet, social isolation, and reduction in physical activity, as well as by dysregulated metabolic and inflammatory pathways [175,176].

#### 3.4.1. Genetic Background

A common genetic basis for psoriasis and obesity was suggested by a study that found a significant correlation between the two conditions in twins [177]. As both diseases have a multifactorial etiology with a polygenic component, several genes have been hypothesized to play a role.

The *FTO* gene encodes a protein implicated in the regulation of the body adipose mass, and its polymorphisms carry a higher risk for obesity and have been associated with a higher PASI in patients with psoriasis [95,178]. Polymorphisms of the *MC4R* gene have been linked to obesity both in the general population and in patients with psoriasis with an increased risk for onset of psoriatic arthritis in the latter [95,179]. As far as genetic variants of adipokines are concerned, no association has been reported between alleles of the leptin [180,181], leptin receptor, adiponectin [181], and omentin [182,183] genes and psoriasis onset or severity.

#### 3.4.2. Molecular Pathways

The adipose tissue is responsible for the active secretion of several bioactive molecules whose effects can spread systemically and play a relevant role in the pathways implicated in psoriatic inflammation [184].

For instance, free fatty acids, which are abundantly released from the adipose tissue of patients with obesity, have been shown to exert a proinflammatory role and to worsen psoriasis in murine models, at least partly via activation of keratinocytes [184]. Moreover, adipokines, cytokines secreted by the adipose tissue, have been demonstrated to interfere with the function of several cell types. Leptin, whose circulating levels are higher in patients with obesity, acts on immune cells, leading to the stimulation of the production of proinflammatory cytokines and reactive oxygen species, chemotaxis, phagocytosis, and proliferation, as well as the polarization of the Th1/Th17 axis in lymphocytes [184,185]. Furthermore, it also stimulates skin resident cells, namely, keratinocytes, fibroblasts, and endothelial cells, further promoting the establishment of a psoriatic milieu [186]. Coherently, murine models genetically deficient for leptin showed attenuation of psoriasis [187], while studies conducted on patients with psoriasis demonstrated higher levels of this hormone compared to healthy controls, even in nonobese subjects [146,188].

Conversely, adiponectin is an adipokine with anti-inflammatory effects whose levels are lower in patients with obesity. In addition to a global suppression of inflammatory cytokine production, adiponectin suppresses IL-17 production in the skin [184,189]. Accordingly, its levels have been demonstrated to be lower in patients with psoriasis [188].

A similar trend has been observed for other adipokines in psoriasis, with an increase in those with proinflammatory effects, including resistin [188,190], chemerin [191], and visfatin [192], and a decrease in those with anti-inflammatory effects, such as omentin [182,183]. Even more significantly, healing of psoriasis following effective therapy results in the normalization of the aberrant levels of these molecules [185,191].

Furthermore, the adipose tissue itself can produce proinflammatory cytokines, such as IL-6, IL-1, and TNF, partly due to a direct secretion from adipocytes and partly from immune cells infiltrating the fat septa [184]. In fat tissues of patients with obesity, the macrophages in the immune cell infiltration show an abnormal polarization toward the M1 phenotype, and lymphocytes show a Th1 shift, both contributing to the enhanced secretion of proinflammatory cytokines [193]. Therefore, in patients with obesity, higher levels of circulating IL-6 [194,195], TNF [194,196], and IL-1 [197] are observed. This abnormal cytokine secretion induces a pathological polarization toward the Th17 axis in obese subjects, as shown by the higher number of IL-17-producing lymphocytes found in both lymphoid and nonlymphoid tissues [198].

Another possible mechanism shared by obesity and psoriasis derives from disturbances in the gut microbiota. Microbial dysbiosis and altered production of bacterial metabolites appear to have strong similarities in the two conditions [199]. Consistently, a study on the composition of human gut microbiota in patients with psoriasis observed significant alterations compared to healthy controls. However, no differences were observed when comparing psoriatic patients with obesity with healthy patients with obesity [200].

Notably, an increased Firmicutes/Bacterioidetes ratio has been associated with both psoriasis and obesity [200]. This imbalance could represent a shared pathogenic mechanism due to the altered production of short-chain fatty acids by these bacterial phyla, which play an important role in metabolic and anti-inflammatory processes [199] Additionally, a decrease in *Akkermansia* species abundance has been reported in both psoriatic patients and patients with obesity, and it may be relevant in light of its role in modulating inflammatory pathways including that of nuclear factor kappa B [200]

### 3.5. NAFLD

An increasing deal of evidence shows that NAFLD is associated with an increased risk of CV events independent of traditional CV risk factors [201]. A 2015 systematic review and meta-analysis indicated that patients with psoriasis have a twofold increased risk of NAFLD compared to controls [29].

In addition to the important role of adipose tissue in mediating the interplay between skin and liver, severe psoriasis may have a direct impact on NAFLD, possibly via mechanisms beyond obesity [202]. Indeed, NAFLD in the general population can also occur among individuals who are not obese and have a normal body mass index (BMI). These individuals are labeled as “lean NAFLD” [203]. Interestingly, subjects with “lean” NAFLD compared to healthy subjects show higher mean serum C-reactive protein levels, suggesting that systemic inflammation might be one of the pathogenic factors of “lean” NAFLD.

Indeed, IL-6, IL-17, and TNF-α produced by the liver (hepatokines) and inflamed skin might have synergistic effects on these organs. This inflammatory-related relationship between psoriasis and NAFLD has been defined as the “hepato-dermal axis” [202]. IL-17 can induce the activation of hepatic stellate cells and subsequent collagen production [204]. By doing so, IL-17 facilitates the progression from simple liver steatosis to steatohepatitis [202,204]. NAFLD to (nonalcoholic steatohepatitis) NASH progression correlates with a higher frequency of IL-17(+) cells among liver CD4(+) T cells and higher Th17/resting Treg and Th2/resting Treg ratios in the blood [205]. Accordingly, IL-17 blockade restores insulin resistance and prevents NAFLD inflammation in mouse models [206]. The genetic ablation of IL-17RA also alleviates NAFLD [206].

## 4. Conclusions

Patients with psoriasis have an increased risk of developing CVD and traditional CV risk factors including hypertension, diabetes, hyperlipidemia, obesity, and NAFLD. The mechanism underlying this association has not been unequivocally defined so far and could be due to shared genetic variants, aberrantly activated molecular pathways, or lifestyle modifications. The literature suggests that psoriasis shares genetic and molecular associations with atherosclerosis and CV risk factors. The nonlinear and intricate interplay among various factors, impacting the molecular pathways in different cell types, probably contributes to the development of psoriasis and CVD. Future research should, therefore, aim to fully unravel shared and differential molecular pathways underpinning the association between psoriasis and CVD. This could enable physicians to customize therapeutic interventions on the basis of individual molecular and clinical profiles.

## Figures and Tables

**Figure 1 ijms-23-09063-f001:**
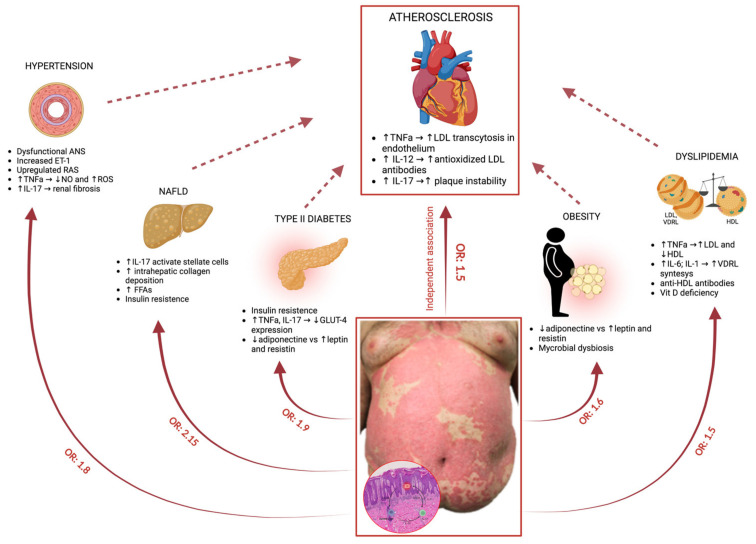
Molecular pathways involved in the link between psoriasis and its comorbidities. Psoriasis *per se* is considered an independent risk factor for atherosclerosis. Moreover, psoriasis is also associated with a high prevalence of traditional modifiable CV risk factors, such as hypertension, diabetes, hyperlipidemia, obesity, and nonalcoholic fatty liver disease (NAFLD) [27,28,29]. Aberrantly activated molecular pathways involved in the link between psoriasis and its comorbidities are indicated.

**Table 1 ijms-23-09063-t001:** List of the shared genetic and molecular pathways between psoriasis and CV events or CV risk factors.

	Atherosclerosis	Hypertension	Diabetes	Hyperlipidemia	Obesity	NAFLD
Shared genetic background with psoriasis	IL-23R and IL-23	-eNOS-LNPEP-AGT	-PSORS2, PSORS3, PSORS4-CDKAL1-JAZF1-ST6GAL1-IL-12B, IL-23R, IL-23A	-HLA gene region-ApoE ^+^-Paraoxonase 1-LXR-α ^++^-PPAR-α ^++^	-FTO *-MC4R ^^^	
Shared molecular pathways with psoriasis	IL-12 *IL-17 *§	-Endothelin-1-Upregulated ACE signaling ^ 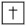 ^-TNFα, IL-17	-TNF-α, IL-23, IL17-adipokines	-TNFα, IL-6 and IL-1β-Dysfunctional HDL and anti-HDL antibodies-Vitamin D deficiency	-Free fatty acids-Adipokines-TNFα, IL-6, and IL-1-Gut microbiota dysregulation	TNF-αIL-17 *°

° IL-17 may promote the progression of NAFLD to NASH; ^+^ both protective and detrimental effects on psoriasis severity depending on the polymorphism; ^++^ downregulated in psoriatic skin; ^^^ associated with psoriatic arthritis; ^
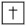
^ increased expression of the renin gene in psoriatic skin indicates a hormone-like action of plaque products; * in patients with higher PASI; § contradicting findings have been reported in different experimental animal models of CVD.

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
