# Peer review of "Psoriasis and Cardiometabolic Diseases: Shared Genetic and Molecular Pathways"

_ijms, 2022, doi:10.3390/ijms23169063_

Round 1
Reviewer 1 Report
Congratulations to the authors for a very thorough analysis of the existing literature with regards to cardiovascular disturbances in psoriasis. A properly constructed manuscript.
1. In my opinion it will be better for understanding add a figure which can explain and sum up – how psoriatic inflammation influences on CVD.
2. Tabl 1 - NAFLD - capital letters
Author Response
Congratulations to the authors for a very thorough analysis of the existing literature with regards to cardiovascular disturbances in psoriasis. A properly constructed manuscript.
Re: we thank the reviewer for the positive feedback.
- In my opinion it will be better for understanding add a figure which can explain and sum up – how psoriatic inflammation influences on CVD.
Re: We thank the reviewer for the suggestion. We now added a figure showing the pathways involved in the link between psoriasis and its comorbidities.
- Tabl 1 - NAFLD - capital letters
Re: It is a general rule to refer to non-alcoholic fatty liver disease as “NAFLD“ with capital letters.
Reviewer 2 Report
The review paper by Stefano Piaserico and co-authors “Psoriasis and cardiometabolic diseases: shared genetic and 2 molecular pathways” is a very good review study on the correlation of skin disease with cardio vascular disease. Some of my concerns for this paper are
Major concerns
1. Please provide a detailed graphical pathway explanation for correlation of incidence of chronic psoriasis with cardiovascular diseases.
2. The manuscript seems to be highly disorganized and the information about TNF alpha, IL12 and IL 7 is mentioned in both section 2 and section 3 in the genetic and molecular background. Please organize them in an appropriate way.
3. In section 3 with the heading “Higher prevalence of traditional CV risk factors in psoriasis patients” the authors have explained the different diseases having shared genetic background with psoriasis such as obesity, diabetes, NAFLD, hypertension and hyperlipidemia, atherosclerosis but failed to connect it to cardio vascular diseases and also did not explain the major types Cardio vascular diseases.
4. I request the authors to please rewrite the major portion of the manuscript and connect all the diseases mentioned with cardio vascular diseases and also create another section about the different cardiovascular diseases.
Minor concerns
1. The introduction can be expanded more
2. Please correct the spacing errors in the reference numbers when indicated in the manuscript. Differences can be observed in Line 156, 165,166 and so forth
3. Why is diabetes included in the section Higher prevalence of traditional CV risk factors in psoriasis patient? It is also a separate chronic disease with high prevalence rate of heart disease but cannot be categorized in cardio vascular disease
Author Response
Major concerns
- Please provide a detailed graphical pathway explanation for correlation of incidence of chronic psoriasis with cardiovascular diseases.
Re: We thank the reviewer for the suggestion, we have included a figure showing the pathways involved in the link between psoriasis and its comorbidities, as required.
- The manuscript seems to be highly disorganized and the information about TNF alpha, IL12 and IL 7 is mentioned in both section 2 and section 3 in the genetic and molecular background. Please organize them in an appropriate way.
Re: The paper was organized in order to describe the shared molecular pathways between psoriasis and cardiometabolic diseases with a structure based on the latter and not to describe the role of the specific molecules. Consequently, a little overlap may exist when explaining the association between psoriasis and different conditions (e.g. with TNF alpha and IL-17).
- In section 3 with the heading “Higher prevalence of traditional CV risk factors in psoriasis patients” the authors have explained the different diseases having shared genetic background with psoriasis such as obesity, diabetes, NAFLD, hypertension and hyperlipidemia, atherosclerosis but failed to connect it to cardio vascular diseases and also did not explain the major types Cardio vascular diseases.
Re: We thank the reviewer for this comment. Indeed, we have described the association of psoriasis with cardiometabolic diseases, namely diabetes, obesity, NAFLD, hypertension and hyperlipidemia. These are all well recognized CV risk factors and therefore implicitly and directly connected with cardiovascular disease. As requested, we have now discussed in the introduction the major types of cardiovascular diseases associated with psoriasis, and specified that the focus or our review is ischemic heart disease.
- I request the authors to please rewrite the major portion of the manuscript and connect all the diseases mentioned with cardio vascular diseases and also create another section about the different cardiovascular diseases.
Re: We thank the reviewer for the suggestion, however, since the aim of the paper, as agreed with the IJMS Editorial Office, was not to discuss all the various cardiovascular diseases, but rather to investigate the pathogenetic link between psoriasis and cardiometabolic diseases, as clearly stated by the title. Therefore, we focused our paper on ischemic heart disease, without including different types of cardiovascular disease as it was not the aim of the review assigned by the Editorial Office of the Journal. We have now specified this in the introduction (“In this paper, we focused our attention on ischemic heart disease (IHD)”).
Minor concerns
- The introduction can be expanded more
Re: We thank the reviewer for the comment. We have now expanded the introduction according to the suggestion.
- Please correct the spacing errors in the reference numbers when indicated in the manuscript. Differences can be observed in Line 156, 165,166 and so forth
Re: We thank the reviewer for this comment, we have corrected the spacing errors.
- Why is diabetes included in the section Higher prevalence of traditional CV risk factors in psoriasis patient? It is also a separate chronic disease with high prevalence rate of heart disease but cannot be categorized in cardio vascular disease
Re: As previously mentioned, the aim of the paper was to discuss the pathogenetic link between psoriasis and cardiometabolic disorders, which include diabetes.
Reviewer 3 Report
Authors should write what abbreviations stay for, also in the abstract, e.g.:
Line 14: write “cardiovascular (CV)” instead of “CV”
Line 21: write “cardiovascular disease (CVD)” instead of “CVD”
Line 67: write “T helper 1 and 17 (Th1 and Th17)” instead of “Th (T helper) 1 and Th17”
Line 104: write “reactive oxygen species (ROS)” instead of “ROS”
Try to put a space between the words and the number of the reference (e.g., lines 30, 32, 36, 39, 55, 68, 80, 89, 90, 91, etc)
Moreover, some words that need to change are:
Line 48: Insert the dot
Line 50: What “(1.5)” stay for? It must be better specified
Line 52: Change “statin therapy” with “therapy with statins”
Line 65: Add “cells” after “dendritic”
Line 75: Eliminate the number 2 above “disease” and add the reference.
Line 103: Change “cardiomyocyte” with “cardiomyocytes”
Line 106: Insert the dot after “[43, 44]”
Line 106: Change “have reported” with “reported”
Line 200: Change “have found” with “found”
Line 213: Remove the comma after “Cheng et al. [77]”
Line 266: Add a comma after “particular”
Line 300: Change “is capable of reducing” with “can reduce”
Line 345: Remove the comma between the words “leptin” and “and resistin”
Line 412: Write “results” instead of “result”
Line 447: Add the space between “the” and “stimulation”
Line 464: Change “is capable of producing” with “can produce”
Line 478: Insert the dot before “however”
The article is well organized in paragraphs and flows to read. Consider adding some images of molecular pathways to make it more understandable.
Author Response
Authors should write what abbreviations stay for, also in the abstract, e.g.:
Line 14: write “cardiovascular (CV)” instead of “CV”
Line 21: write “cardiovascular disease (CVD)” instead of “CVD”
Line 67: write “T helper 1 and 17 (Th1 and Th17)” instead of “Th (T helper) 1 and Th17”
Line 104: write “reactive oxygen species (ROS)” instead of “ROS”
Try to put a space between the words and the number of the reference (e.g., lines 30, 32, 36, 39, 55, 68, 80, 89, 90, 91, etc)
Moreover, some words that need to change are:
Line 48: Insert the dot
Line 50: What “(1.5)” stay for? It must be better specified
Line 52: Change “statin therapy” with “therapy with statins”
Line 65: Add “cells” after “dendritic”
Line 75: Eliminate the number 2 above “disease” and add the reference.
Line 103: Change “cardiomyocyte” with “cardiomyocytes”
Line 106: Insert the dot after “[43, 44]”
Line 106: Change “have reported” with “reported”
Line 200: Change “have found” with “found”
Line 213: Remove the comma after “Cheng et al. [77]”
Line 266: Add a comma after “particular”
Line 300: Change “is capable of reducing” with “can reduce”
Line 345: Remove the comma between the words “leptin” and “and resistin”
Line 412: Write “results” instead of “result”
Line 447: Add the space between “the” and “stimulation”
Line 464: Change “is capable of producing” with “can produce”
Line 478: Insert the dot before “however”
Re: We thank the reviewer for these suggestions, and we have amended the text accordingly.
The article is well organized in paragraphs and flows to read. Consider adding some images of molecular pathways to make it more understandable.
Re: We thank the reviewer for the comment. We have added a graphical description of molecular pathways as suggested.
Round 2
Reviewer 2 Report
The authors have answered my queries. Although I still feel the need to do some minor revision work and explain more about ischemic heart disease (IHD), Besides that, I have no other questions. Thank you.
Author Response
The authors have answered my queries. Although I still feel the need to do some minor revision work and explain more about ischemic heart disease (IHD), Besides that, I have no other questions. Thank you.
Re: We thank the reviewer for the positive comment. As suggested, we have added an explanation about ischemic heart disease (IHD) in the introduction section.